# Predictors of rapid eye movement sleep behavior disorder in patients with Parkinson's disease based on random forest and decision tree

**Wu Chong-Wen, Li Sha-Sha, E. Xu** *

Department of Medical, Huzhou Normal University, Huzhou, Zhejiang Province, China

* 02506@zjhu.edu.cn

## Abstract

### Background and objectives

Sleep disorders related to Parkinson's disease (PD) have recently attracted increasing attention, but there are few clinical reports on the correlation of Parkinson's disease patients with rapid eye movement (REM) sleep behavior disorder (RBD). Therefore, this study conducted a cognitive function examination for Parkinson's disease patients and discussed the application effect of three algorithms in the screening of influencing factors and risk prediction effects.

### Methods

Three algorithms (logistic regression, machine learning-based regression trees and random forest) were used to establish a prediction model for PD-RBD patients, and the application effects of the three algorithms in the screening of influencing factors and the risk prediction of PD-RBD were discussed.

### Results

The subjects included 169 patients with Parkinson's disease (Parkinson's disease with RBD [PD-RBD] = 69 subjects; Parkinson's disease without RBD [PD-nRBD] = 100 subjects). This study compared the predictive performance of RF, decision tree and logistic regression, selected a final model with the best model performance and proposed the importance of variables in the final model. After the analysis, the accuracy of RF (83.05%) was better than that of the other models (decision tree = 75.10%, logistic regression = 71.62%). PQSI, Scopa-AUT score, MoCA score, MMSE score, AGE, LEDD, PD-course, UPDRS total score, ESS score, NMSQ, disease type, RLSRS, HAMD, UPDRS III and PDOnsetage are the main variables for predicting RBD, along with increased weight. Among them, PQSI is the most important factor. The prediction model of Parkinson's disease RBD that was established in this study will help in screening out predictive factors and in providing a reference for the prognosis and preventive treatment of PD-RBD patients.

**Data Availability Statement:** All relevant data are within the paper.

**Funding:** The author(s) received no specific funding for this work.

**Competing interests:** The authors have declared that no competing interests exist.

## Conclusions

The random forest model had good performance in the prediction and evaluation of PD-RBD influencing factors and was superior to decision tree and traditional logistic regression models in many aspects, which can provide a reference for the prognosis and preventive treatment of PD-RBD patients.

## Introduction

Sleep disorder is a type of nonmotor symptom that afflicts almost all PD patients. In addition to its own influence on patients' moods, quality of life and social function, sleep disorders can interact with other types of nonmotor symptoms, such as depression and cognitive decline. Thus, the loss of daily living activities in PD patients is accelerated [1]. Among them, approximately 1/3 of patients exhibit rapid eye movement REM sleep behavior disorder (RBD) [2]. It is noteworthy that RBD does not occur in all PD patients, and some clinical observations have shown that affected PD patients may have specific clinical manifestations. RBD can also be a premonitory symptom of PD [3]. Currently, the understanding of PD-RBD is relatively limited; thus, it is necessary to conduct clinical studies on RBD in PD patients. Specifically, we need to fully understand the scope of the influence of RBD on PD, as well as its related factors and its relationship with PD progression.

The pathogenesis of PD-RBD is complex and unclear. Existing studies have mostly focused on the verification and determination of influencing factors of PD-RBD; for example, risk factors for PD-RBD are evaluated by scale, and the risk of a factor on PD-RBD is determined by performing a logistic regression analysis via ORs [4]. There are few studies concerning the risk assessment of PD with RBD under the comprehensive effect of multiple factors, and traditional statistical methods are mostly used to explore the individual risk of PD-RBD from biological, psychological and social dimensions, which are prone to problems of overfitting and low prediction accuracy and validity [5–7]. Therefore, the question of how to integrate biological, psychological, social and other multidimensional factors to build a PD-RBD risk prediction model, as well as to screen out the key influencing factors of PD-RBD and to effectively distinguish the risk of PD with RBD, is the key to RBD risk prediction research.

Random forest (RF) is a classification algorithm that was proposed by Leo Breiman [8]. It is a representative set-based machine learning technique that improves the accuracy and stability of predictions by using multiple decision trees. In recent years, the decision tree algorithm has been widely used in disease risk prediction, early warning and prognosis. It can effectively process mixed data, missing values or outlier values, as well as higher-dimensional data in medical data; additionally, it is not prone to overfitting, and the prediction accuracy is relatively accurate [9]. Studies [10] have applied random forest and decision tree algorithms in genomic research, classification and the discrimination of individual diseases of schizophrenia and depression, among other disorders, and the results showed that random forest and decision tree algorithms have great advantages in screening important metabolites and in effectively discriminating individual diseases. PD, schizophrenia and depression, among other diseases, are caused by the comprehensive action of multiple factors. Although RF is known to have a higher predictive performance than decision trees [11], prediction studies using biomarkers [12] and images [13] are mainly used to evaluate diseases. In addition, only a few RF-based studies use questionnaire data of social demographic factors, health habits or neuropsychological examination data [14]. However, at present, random forest and decision tree are used for

the risk prediction of RBD, and there are few studies on the quantitative analysis of prevalence and the search for key influencing factors. Therefore, this study established a random forest model for PD-RBD risk prediction by collecting multidimensional factors such as demographic economics, biology, psychosociology, Parkinson's nonmotor symptom score, sleep correlation score, drug dose for Parkinson's treatment and other characteristics. Additionally, this study explored the application effect of the random forest algorithm in the screening of PD-RBD influencing factors and risk prediction.

Therefore, in this study, we collected multidimensional factors such as demographic economics, biology, psychosociology, Parkinson's nonmotor symptom score, sleep correlation score, drug dosage for Parkinson's treatment and other characteristics. The random forest model of PD-RBD risk prediction was established by using logistic regression, machine learning-based regression trees and random forest algorithms, and the application effect of the random forest algorithm in the screening of influencing factors and risk prediction research of PD-RBD was discussed.

## Materials and methods

### Participants

A total of 169 patients with Parkinson's disease who were admitted to the Department of Neurology, The First Affiliated Hospital of Huzhou Normal University from April 2019 to January 2021 were recruited, and demographic data, including sex, age, onset age, course of disease, education level and onset type, were recorded. All of the subjects signed informed consent forms. The occurrence, development and related factors of PD-RBD in 169 patients were reviewed and prospectively analyzed.

The inclusion criteria for Parkinson's disease included patients who met the diagnostic criteria for primary Parkinson's disease, as established by the British Brain Bank in 1992, including (1) bradykinesia (reduced amplitude of voluntary movement, progressive language and repetitive movements) and (2) at least one of the following symptoms: myotonia, 4–6 HZ static tremor and postural instability (not caused by vision, vestibular function, cerebellar or proprioceptive disturbances).

The exclusion criteria for Parkinson's disease included stepped-up Parkinson-like symptoms caused by repeated stroke history, a history of repeated head trauma, movement-eye crisis, a clear history of encephalitis, a history of sedative medication prior to symptom onset, more than 1 sick relative, continued remission of symptoms and severe unilateral symptoms still present after 3 years. Other criteria included supranuclear gaze paralysis, cerebellar symptoms, early onset of severe autonomic nervous dysfunction, early onset of severe dementia, memory, language and behavior abnormalities and the Babinski sign being positive. Additionally, other exclusion criteria included head CT detecting cerebellar tumor or traffic hydrocephalus, no response to high-dose levodopa treatment (excluding malabsorption) and a contact history of 1-methyl 4-stupyl-1,2,3,6-tetrahydropyridine.

The following supporting diagnostic criteria were used for the diagnosis of Parkinson's disease (3 or more are required for the diagnosis of Parkinson's disease): unilateral onset, static tremors, progressive course of the disease, long-term asymmetry of symptoms, the onset of obvious symptoms, good response to levodopa (70–100%), the appearance of levodopa-induced chorea, reaction to levodopa for 5 years or more and a clinical course of the disease being 10 years or more. All of the subjects signed informed consent forms.

## Clinical assessment

RBDSQ was used to screen patients with RBD for Parkinson's disease. RBDSQ was developed in 2007 by Stiasny-Kolster and other doctors from The University of Marburg in Germany based on the diagnostic criteria for RBD in the second edition of the International Classification of Sleep Disorders, encompassing the clinical characteristics of RBD. The results were consistent with the gold standard polysomnogaphy (PSG) for RBD diagnosis. RBDSQ consists of 10 items. The first item is the frequency and content of dreams and their relationship with nocturnal behavior, the fifth item is whether nocturnal behavior harms patients and bed partners and the sixth item contains 4 subitems to evaluate subjects' nocturnal behavior, including shouting, body movements and falling into bed. Items 7 and 8 relate to nocturnal arousal. Item 9 involves overall sleep status. The 10th item involves whether the patient had neurological diseases. The total score can equal 13 points. The specificity and sensitivity of ≥6 were 0.96 and 0.84, respectively, for the positive boundary value [1]. In this study, PD-RBD was ≥6. The partner was encouraged to assist in the evaluation, with "yes" being given 1 point and "no" being given 0 points.

## Subject questionnaires

Evaluation of motor function. (1) Motor Function(MS): the patients' motor function symptoms were evaluated by using the Unified Parkinson's Disease Rating Scale III (UPDRS III) [15] and Hoehn-Yahr (H-Y) stages [16]; and (2) motor complications: UPDRS IV [15] was used to evaluate dyskinesia, and the number of wearing-off symptoms was evaluated by using the wearing-off symptom scale.

Evaluation of NMS. (1) NMS screening: the Nonmotor Symptom Quest (NMSQ) [17] was used to screen NMS, and the total number of NMS and the number of NMS before and after the MS stage were counted; (2) overall mental and emotional status: UPDRS I [15] was used to evaluate the overall mental and emotional statuses of the patients; (3) cognitive function: the Montreal Cognitive Assessment Scale (MoCA) [18] and Mini-Mental State Examination (MMSE) [19] were used to evaluate the cognitive function of patients; (4) depression: the 24-item version of the Hamilton Depression Scale (HAMD) [20] was used to evaluate the patients' depression symptoms; (5) anxiety: the 14-item Version of the Hamilton Anxiety Scale (HAMA) [21] was used to evaluate the anxiety statuses of the patients; (6) degree of daytime sleepiness degree: the Epworth Sleeping Scale (ESS) [22] was used to evaluate the daytime sleepiness degree of patients; (7) overall sleep: the Pittsburgh Sleep Quality Index (PSQI) [23] was used to evaluate the overall sleep of the patients; (8) autonomic function: the Scale for Outcomes in PD for Autonomic Symptoms (SCOPA-AUT) [24] was used to evaluate the autonomic symptoms; (9) Restless Leg Syndrome(RLS): RLS and its manifestations and severity were evaluated by using the Restless Leg Syndrome Rating Scale (RLSRS) [25]; and (10) the Nonmotor Symptom Questionnaire (NMSQ) [26] was used to evaluate gastrointestinal digestive function, autonomic nervous symptoms, neuropsychiatric symptoms, sleep disorders and sensory disorders.

Evaluation of drug use. The following calculation formula of levodopa equivalent daily dose (LEDD) was used: LEDD = Levodopa controlled-release tablets×0.75+Levodopa controlled-release tablets×0.75×0.25 (while taking entacapone tablets at the same time)+Pramipexole hydrochloride tablets×100+pibedil sustained-release tablets×L+Selegiline hydrochloride tablets×10+Rasagiline mesylate tablets×100+ amantadine×1, in mg/day.

### Ethics statement

All participants provided their written consent. The study was approved by the institutional review board at Huzhou Normal University which is in accordance with the Declaration of Helsinki.

### Statistical analysis

**Data processing.**   Data processing represents the most important link for business data to be transformed into research data. The following steps were performed in this study. The first step involved data proofreading, which corrects obvious data errors by tracing the source; the second step involved data combination. A reasonable combination of outpatient and inpatient patients was performed, and a total of 169 patients were enrolled. The third step involved the processing of missing values. Due to the fact that there is no field of specimen source in the original data, the field should be completed through the business system and professional judgment, and the entries that cannot be completed should be deleted. The fourth step is to determine the meaning of the outliers and to convert the classification variables of the outliers into computable numerical variables.

**Model establishment.**   In this study, stratified sampling was used to select 70% of the samples from the PD-RBD cases and PD-nRBD cases to form a training set for building the model, and the remaining 30% of the samples were used as test sets to evaluate the performance of the model. Twenty-one characteristic variables were selected as the input variables, and the occurrence of RBD was used as the outcome variable. Random forest, logistic regression and decision tree models were established in the training set. Finally, each model was used to evaluate the prognosis of upper gastrointestinal bleeding in the test dataset, and $P \leq 0.05$ was considered to be statistically significant.

The logistic regression model was established by using the training set under the GLM function to build a logistic model. The Step function was used to select the stepwise regression variables based on the AIC for the constructed initial logistic model.

To establish the decision tree model, the following parameters were used in this study to establish the initial decision tree. The minimum sample size parameter minbucket of the tree0 leaf node was set to 20; the minimum sample size parameter minsplit of nodes was set to 20. The pruning number parameter xval for cross-validation was set to 10. The maximum depth parameter maxdepth of decision tree generation was set to 20. This specifies that the complexity parameter CP in the minimum cost complexity pruning was set to 0.01.

In the R language environment, the modeling process of random forest mainly contains two important numbers: Ntree (number of trees) and Mtry (number of randomly selected features). It has been proven that the random forest model performs best when the parameter ntree is set to 500 and mtry is set to 4.

**Data analysis.**   Relevant data were extracted by using the HIS and LIMS systems, and the data were exported to Excel. The decision tree model and random forest model were established via R language for model establishment and evaluation. The statistical function package was called by R language for batch data processing, which was verified by using SPSS 21.0 statistical software. Count data are represented as an example (%). Logistic regression was used to analyze the correlation between the data and whether patients had PD-RBD. The receiver operating characteristic curve (ROC curve) was drawn to evaluate the prognostic effect of the RBD prediction model on PD patients.

## Results

### Demographic and clinical characteristics of PD-RBD and PD-nRBD

Among 169 patients with Parkinson's disease, 69 patients had RBD (PD-RBD), and the incidence was 40.83%. One hundred patients without RBD (PD-nRBD) accounted for 59.17%. Among 169 patients with PD, 43 cases (25.44%) were from the neurodegenerative disease ward of the neurology department, and 126 cases (74.55%) were from the outpatient department. Eighty-four patients were male (49.70%), and 85 patients were female (50.30%). The age range was 54 to 71 years, with a mean age of 64.07±5.24 years. The onset age ranged from 46 to 70 years, with an average age of 59.20±5.98 years. A total of 109 cases (64.50%) had primary school education, 44 cases (26.04%) had secondary school education and 16 cases (9.47%) had junior college education. The course of the disease ranged from 1 to 10 years. The onset types of PD-RBD were as follows: 20 cases of rigidity, 35 cases of tremor, 7 cases of abnormal gait and 7 cases of mixed type. The onset types of PD-nRBD were as follows: 20 cases of rigidity, 31 cases of tremor, 23 cases of abnormal gait and 26 cases of mixed type.

The comparison of the demographic data between the PD-RBD group and the PD-nRBD group showed that the course of the disease in the PD-RBD group was significantly longer than that in the PD-nRBD group. There were no significant differences in sex, age, onset age or education level between the two groups, and there were significant differences in onset types (rigidity, tremor, abnormal gait and mixed type).

Motor symptoms. The H-Y staging of the PD-RBD group was significantly higher than that of the PD-nRBD group, and the UPDRS III motor symptom scores of the two groups were significantly different. There was no significant difference between the PD-RBD group and the PD-nRBD group in regards to UPDRS III score or the number of symptoms at the end of treatment.

NMS score. MoCA, MMSE, ESS, PQSI, NMSQ, SCOPA-AUT, UPDRS, UPDRS III score and LEDD dose were significantly higher in the PD-RBD group than in the PD-nRBD group. The details are shown in Table 1.

### Development results of the PD-RBD prediction model based on the logistic regression model

The results of the univariate analysis of whether PD patients were complicated with RBD included MoCA, MMSE, ESS, PQSI, NMSQ, SCOPA-AUT, UPDRS, UPDRS III exercise score, LEDD dose, H-Y stage, disease onset type and PD course.

Before model training, the independent variables were screened by using logistic regression. In the training dataset, a logistic regression model was established concerning whether PD patients were complicated with RBD as the dependent variable and 12 indicators of the collected data as independent variables, and the test level was set at 0.05. Specific results are shown in Table 2.

### Development results of the PD-RBD prediction model based on the decision tree model

In this study, the decision tree model selects the information state as the metric of the split attribute.

In the process of the prepruning of the decision tree, the prepruning parameters mainly included the maxdepth, Minbucket and minsplit. In the verification dataset, different prepruning parameters were selected to build the decision tree model. It was found that changing the prepruning parameters had little influence on the accuracy of the decision tree model. This

**Table 1. Demographic and clinical characteristics of patients.**

| Item | PD-RBD(n = 69) | PD-nRBD(n = 100) | $x^2$/t | P |
|---|---|---|---|---|
| **Female: Male** | 37:32 | 48:52 | 0.516 | 0.472 |
| **Age** | 64.4±4.7 | 63.8±5.6 | 0.761 | 0.448 |
| **Education Level** | | | 0.083 | 0.959 |
| Primary and below | 45 | 64 | | |
| secondary school | 18 | 26 | | |
| Junior college or above | 6 | 10 | | |
| **Disease type** | | | 14.517 | 0.002* |
| Less rigid motion | 20 | 20 | | |
| tremor | 35 | 31 | | |
| Abnormal gait | 7 | 23 | | |
| hybrid | 7 | 26 | | |
| **Age of onset of PD (years)** | 58.9±5.7 | 59.4±6.2 | -0.561 | 0.576 |
| **PD duration (years)** | 5.6±2.4 | 4.4±1.9 | 3.277 | 0.001* |
| **UPDRS** | 36.7±10.0 | 29.7±9.0 | 2.050 | 0.042* |
| **UPDRS III movement** | 18.4±2.6 | 17.2±2.7 | 2.893 | 0.004* |
| **UPDRS I spirit** | 4.4±1.9 | 4.2±2.1 | 0.427 | 0.670 |
| **SCOPA-AUT** | 34.3±4.6 | 32.3±4.2 | 3.004 | 0.003* |
| **MoCA** | 15.9±3.5 | 17.4±3.8 | -2.543 | 0.012* |
| **LEDD** | 564.3±210.4 | 504.2±146.6 | 2.051 | 0.043* |
| **ESS** | 6.1±1.6 | 5.3±1.5 | 3.240 | 0.001* |
| **PQSI** | 8.5±3.8 | 6.9±3.5 | 3.020 | 0.003* |
| **NMSQ** | 8.9±3.8 | 7.3±3.5 | 2.875 | 0.005* |
| **HY stage** | | | 7.823 | 0.020* |
| 1–1.5 | 15 | 40 | | |
| 2–3 | 27 | 37 | | |
| 4–5 | 27 | 23 | | |
| **MMSE** | 24.0±3.5 | 22.5±4.2 | 2.512 | 0.013* |
| **RLSRS** | 6.0±4.6 | 4.9±5.0 | 1.420 | 0.157 |
| **HAMA** | 4.6±1.6 | 4.3±1.6 | 1.293 | 0.198 |
| **HAMD** | 7.6±3.1 | 7.5±3.0 | 0.364 | 0.716 |
| **Dyskinesia** | | | 0.049 | 0.826 |
| yes | 35 | 49 | | |
| no | 34 | 51 | | |
| **End of dose phenomenon** | | | 1.458 | 0.227 |
| yes | 1 | 0 | | |
| no | 68 | 100 | | |

study mainly selected the minimum cost complexity parameter CP in the process of rear pruning of the decision tree. The specific results are shown in Table 3.

In the selection principle of the CP value, the prediction error and standard error of the model are usually considered. As seen from Table 3, when the number of splits is 4, the prediction error of the model is at least 0.6739; thus, a decision tree model tree_CP 1 with a CP value of 0.0435 is established. When the number of splits is 2, the number of tree nodes in the decision tree model is 3 (the number of splits+1), and the complexity of the tree model is the lowest among all of the models. Due to the fact that the CP value must be between 0.1086 and 0.1956 (CP value is selected as 0. 15 in this example), a decision tree model was set with a CP value of

**Table 2. Results of the logistic regression model.**

| Indicators | β | SE | OR | 95%CI | P |
|---|---|---|---|---|---|
| Disease type | -0.574 | 0.218 | 0.563 | 0.367–0.863 | 0.008* |
| PD duration (years) | 0.286 | 0.108 | 1.331 | 1.077–1.644 | 0.008* |
| UPDRS | 0.012 | 0.023 | 1.012 | 0.968–1.059 | 0.595 |
| UPDRIII movement | 0.130 | 0.078 | 1.139 | 0.977–1.328 | 0.096 |
| SCOPA-AUT | 0.117 | 0.048 | 1.124 | 1.024–1.234 | 0.014* |
| MoCA | -0.101 | 0.058 | 0.904 | 0.807–1.012 | 0.079 |
| LEDD | 0.000 | 0.001 | 1.000 | 0.998–1.003 | 0.748 |
| ESS | 0.388 | 0.141 | 1.474 | 1.119–1.942 | 0.006* |
| PQSI | 0.173 | 0.068 | 1.189 | 1.041–1.358 | 0.011* |
| NMSQ | 0.104 | 0.060 | 1.110 | 0.986–1.250 | 0.084 |
| HY stage | 0.600 | 0.268 | 1.822 | 1.078–3.081 | 0.025* |
| MMSE | 0.125 | 0.058 | 1.134 | 1.012–1.270 | 0.030* |

0. At the same time, a conventional tree model tree_cp3 without minimal cost complexity pruning is established. The results showed that the sum of the sensitivity and specificity of TREE_CP1 is the largest.

The sensitivity of the obtained model is low when modeling under the above parameters; therefore, we introduced the penalty factor into the model (through the loss matrix). In this study, loss matrices of 1, 1.5, 2, 2.5 and 4 times the cost were set. The c (0,3,1,0) loss matrix was verified by the test dataset to have the highest sensitivity (73.6%). The visualization of this is shown in Fig 1.

## Development results of the PD-RBD prediction model based on the random forest model

The number of trees was set to 100, 200, 300, 400 and 500. Through validation dataset verification, the prediction performance of the random forest models with different tree numbers was compared. It was found that when ntree = 170 and Mtry = 7, all of the indicators of the model tended to be stable. Therefore, we chose ntree = 170 and mtry = 7 for the modeling.

The results showed that, as shown in Fig 4, PQSI, SCOPA-AUT, MoCA, MMSE, AGE, LEDD, PD-Course, UPDRS, ESS, NMSQ, disease type, RLSRS, HAMD, UPDRS III, PD-onset age and other indicators have a great influence on the accuracy of the random forest model. These indicators may have great clinical significance (as seen in Fig 2).

**Table 3. Prediction error estimation and complexity parameter CP value of the decision tree model with different splitting times.**

| nsplit | rel error | xerror | Xstd | CP |
|---|---|---|---|---|
| 0 | 1.0000 | 1.0000 | 0.1151 | 0.1956 |
| 1 | 0.8043 | 0.9782 | 0.1147 | 0.1521 |
| 2 | 0.6521 | 0.8478 | 0.1110 | 0.1086 |
| 4 | 0.4348 | 0.6739 | 0.1039 | 0.0435 |
| 5 | 0.3913 | 0.7826 | 0.1087 | 0.0109 |
| 7 | 0.3695 | 0.7826 | 0.1087 | 0.0100 |

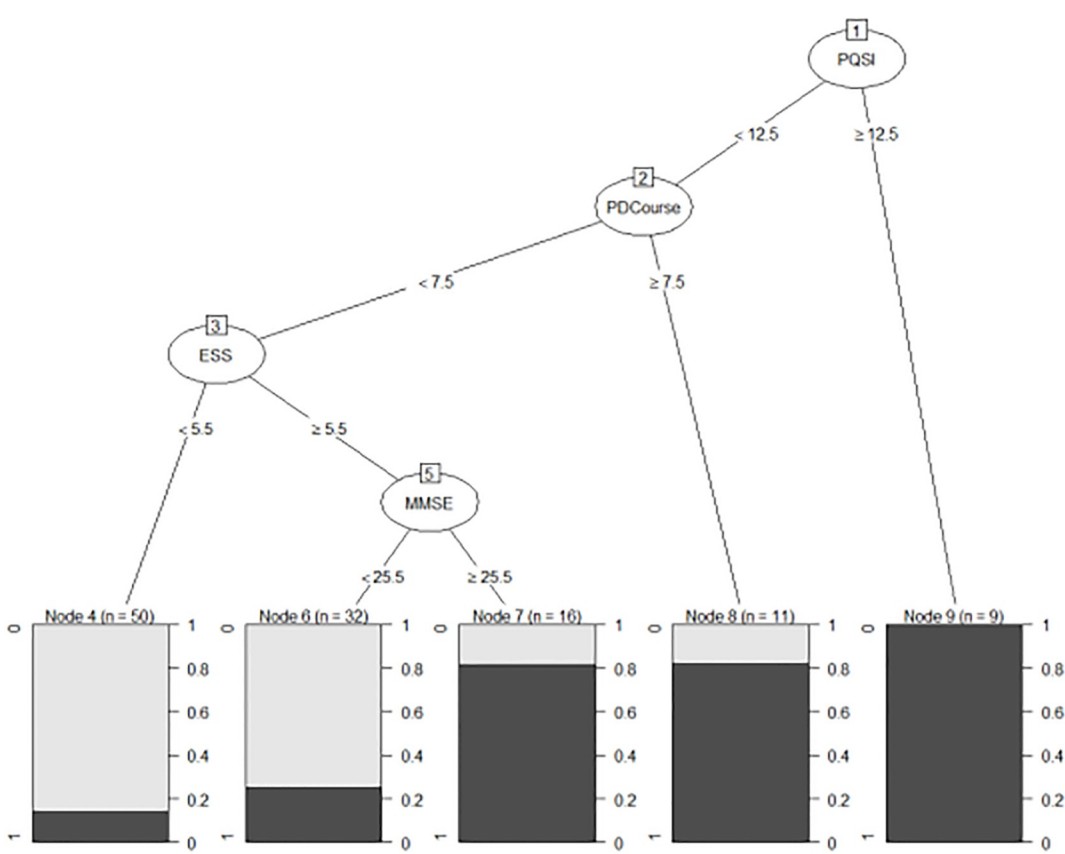

**Fig 1. Visualization of decision tree model.**

## Comparison of the performance of the three models

The final logistic regression model, random forest model and decision tree model were tested in the test dataset, and the test results were compared with the real situation, as shown in Table 4.

In the test set, the random forest model performed better than the other two models in terms of sensitivity, specificity, positive prediction rate, negative prediction rate, accuracy and AUC. Moreover, the highest AUC of the random forest was 80.73%. In the ROC curve, a greater area under the curve corresponded to a greater AUC VALUE, thus indicating better model performance. In summary, the random forest model had good performance in the prediction and evaluation of PD-RBD influencing factors and was superior to decision tree and traditional logistic regression models in many aspects, which can provide a reference for the prognosis and preventive treatment of PD-RBD patients (as seen in Figs 3 and 4).

## Discussion

Early screening for RBD is important in the health sciences because RBD is the most common nonmotor symptom in PD, and it is highly likely to affect the quality of life of patients and caregivers. According to the description of the International Classification of Sleep Disorders, RBD is characterized by intermittent enhancement of EMG activity during REM sleep, as well as complex activities and abnormal behaviors related to dream content [2]. RBD is a risk factor

**Variable Importance Plot - PDRBD**

**Fig 2. Importance of random forest model to predict independent variables.**

and precursor of neurodegenerative diseases, especially Parkinson's disease and other synu-cleinopathies [3]. The risk of developing Parkinson's disease in patients with RBD is 20–45% within 5 years and 40–65% within 10 years [5]. PSG is the gold standard for the diagnosis of RBD, but it is time-consuming and arduous to perform. Screening for suspected RBD with an appropriate scale can help to identify patients who requires further diagnoses and interventions as early as possible. RBDSQ is one of the most suitable tools that are widely used for screening patients with Parkinson's disease associated with RBD, with a high sensitivity and specificity [27]. This tool can save time and effort, as well as reduce the economic burden on patients' families and society. Through the uses of questionnaires or interviews, it was found

**Table 4. Comparison of regression trees, random forest and logistic model performance in the training set and test set.**

| Index | Regression trees | | Random Forest | | Logistic | |
|---|---|---|---|---|---|---|
| | Trainset | Testset | Trainset | Testset | Trainset | Testset |
| Sensitivity/% | 67.39 | 56.52 | 100 | 67.39 | 69.57 | 60.87 |
| Specificity/% | 93.06 | 82.14 | 100 | 93.06 | 86.11 | 71.43 |
| Positive prediction rate/% | 86.11 | 72.22 | 100 | 86.11 | 76.19 | 63.64 |
| Negative prediction rate/% | 81.71 | 69.70 | 100 | 81.71 | 81.58 | 68.97 |
| Accuracy/% | 83.05 | 70.59 | 100 | 83.05 | 79.66 | 66.67 |
| AUC/% | 92.24 | 75.10 | 100 | 80.73 | 90.01 | 71.62 |

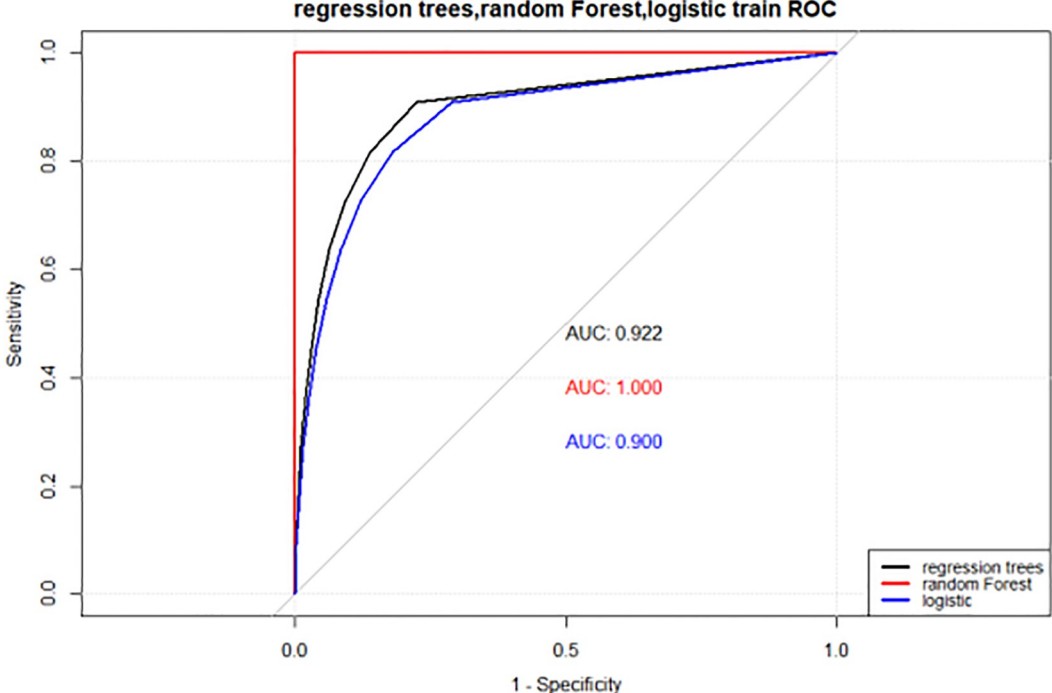

**Fig 3. ROC curves of logistic, decision tree and random forest models in the training set.**

that the incidence of RBD in PD patients was 15–47% [28], and 33% of PD patients were found to have RBD through PSG [29]. The result of this study was 40.83%, which was similar to what has been previously reported. Siegel et al. found for the first time in a cat RBD model that damage to the area adjacent to the bilateral pons can eliminate muscle dystonia, walking

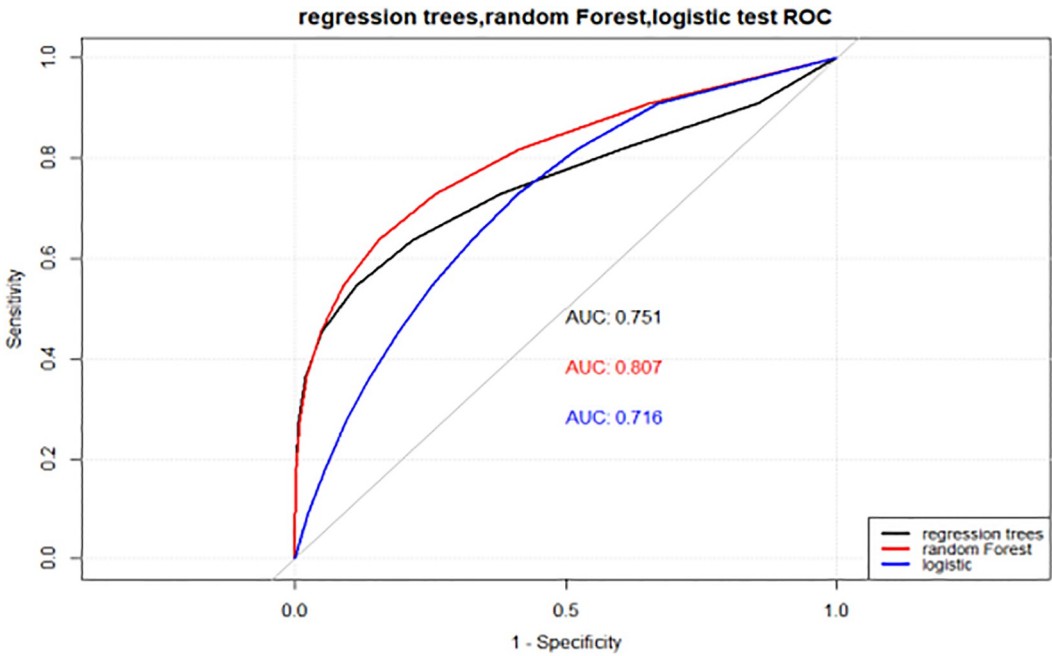

**Fig 4. ROC curves of logistic, decision tree and random forest models in the test set.**

and other complex abnormal behaviors during REM sleep, and this phenomenon has since been found in other animals [30]. It is believed that RBD may be related to the degeneration of dopaminergic neuron pathways in the ventral and dorsolateral tegmental nuclei at the midbrain and pons junction, as well as in the interpodal tegmental nuclei, locus coeruleus, sublocus coeruleus and surrounding areas and the midbrain striatum [31]. N-methyl-d-aspartic acid was injected into the ventral midbrain region of cats, and insomnia and frequent awakenings were observed, followed by Parkinson-like tremors, myotonia and immobility at one week later. The anatomical proximity of these regions links Parkinson's disease to RBD.

In this study, the key discriminative indices of RBD based on RF were discussed while considering sociodemographic factors, evaluations of motor function, evaluations of nonmotor symptoms and evaluations of medications. The PQSI, SCOPA-AUT score, MoCA score, MMSE score, AGE, LEDD, PD course, UPDRS total score, ESS score, NMSQ, disease type, and RLSRS, HAMD, UPDRS III and PD onset age were found to be the key predictors.

Among them, PQSI is the most important predictor, and ESS score is the key predictor. This study found that the overall sleep quality of the RBD group was poor, daytime excessive sleep was prominent and autonomic nervous dysfunction was serious. The PQSI of the PD-RBD group was significantly lower than that of the PD-nRBD group. RBD patients have more dreams at night and exhibit more abnormal behaviors, such as body movement and shouting, thus resulting in sleep interruption and decreased sleep quality. Furthermore, autonomic symptoms, such as increased nocturia and sweating disorders, are prominent, which can further reduce the sleep quality of RBD patients, thus affecting daytime activities and awakening, which makes these patients more prone to daytime oversleeping [2]. Daytime oversleeping is significantly and positively correlated with the course of disease and autonomic nervous dysfunction [32].

Among them, age, PD course, PD onset age and disease type are important predictors. This study found that the course of the disease in patients with PD-RBD is significantly longer than that in patients with PD-nRBD, which is consistent with the results of another study [33]. Patients with RBD have a longer course of disease. Moreover, Lewy body deposition leads to more extensive damage to dopaminergic neurons. Lee et al. [34] reported that via a logistic regression analysis, older age, long duration of PD, onset time of PD, severe degree of disability, type of tremor in disease type and increases in tremor scores were significant risk predictors of RBD in PD patients.

This study found that, compared with the PD-nRBD group, the PD-RBD group had a higher H-Y, thus indicating a more severe degree of disease; thus, the dosage of the dopaminergic drug LEDD was higher, and the exercise complications were more severe. In this study, based on the RF model, the UPDRS III score was found to be a predictor, but there was no significant correlation between the UPDRS III scores of the two groups according to the logistic regression multifactor model. UPDRS III is a subtle evaluation of patients' motor symptoms. In this study, patients in the two groups were mostly observed to be in the mild stage of the disease (the average H-Y stage was stage 1.5 and stage 2.25 for the 2 groups). Therefore, drugs could optimally improve motor symptoms, thus leading to no significant difference in UPDRS III scores between the two groups. However, H-Y staging is divided according to the side of the symptoms that are involved, whether the trunk is involved or not and its severity. The disease requires a certain period of time to reach a certain stage; therefore, it is difficult for drug treatment to significantly improve the H-Y staging.

The incidence of nonmotor symptom NMS in the PD-RBD group was significantly higher than that in the PD-nRBD group, especially the score of the Nonmotor Symptom Questionnaire (NMSQ), which was significantly higher than that in the PD-nRBD group, thus

suggesting that Parkinson's patients with RBD have more extensive central nervous system degeneration [35].

The SCOPA-AUT scores of patients with PD-RBD were significantly higher than those of the PD-nRBD group, thus indicating more severe autonomic neurological symptoms. Postum et al. [36] found that patients with RBD have more frequent autonomic nervous function symptoms, including constipation, urinary symptoms and male sexual dysfunction, which may be related to the overlap of RBD-related regions, such as the locus coeruleus and pontopedral nucleus [37]. For example, histopathology has found that the subprachiuleuprechal complex is associated with REM sleep in patients with RBD, and damage to this region can also lead to cardiac autonomic nervous dysfunction.

Based on the RF predictive model, this study found that anxiety is a predictive factor. Although Parkinson's disease patients exhibit a stress response and psychological pressure to the disease, which results in anxiety, patients with RBD can further aggravate anxiety by having their sleep quality become affected. In PD patients, the pons locus coeruleus, raphe nucleus and substantia nigra are involved, and the levels of norepinephrine, serotonergic and dopaminergic neurotransmitters projected to the orbital surface of the frontal lobe and caudate nucleus are decreased, thus resulting in anxiety. The reductions of these neurotransmitters are also closely related to anxiety. Damage to the locus coeruleus and raphe nucleus also leads to RBD, which establishes an anatomical and pathological link between anxiety and RBD. Moreover, anxiety symptoms themselves also exist, such as difficulty falling asleep and early awakening. RBD in Parkinson's disease patients and anxiety are mutually causal and mutually promoting, thus resulting in a vicious cycle. Sixel-doring et al. [38] also reported that psychiatric comorbidities, such as anxiety and depression, were significantly associated with RBD.

Based on the RF prediction model, this study found that the MMSE and MoCA scores of the PD-RBD group and the PD-nRBD group were of higher importance. The multifactor logistic regression model and decision tree also found that the two groups had a significant correlation in MMSE scores, which was the same result as the previous research finding that RBD patients were mostly complicated with cognitive dysfunction [39]. More patients with Parkinson's disease combined with RBD exhibit mild cognitive impairment (MCI) [7, 39, 40], mainly in the areas of visual space construction, executive function and word memory impairment. Other studies have found that [41] the decision-making ability and feedback learning ability of RBD patients are significantly reduced in a complex environment, and other cognitive areas are relatively late affected; thus, these special cognitive areas are often ignored. Therefore, a more targeted scale should be used to evaluate the cognitive function of patients with PD-RBD in clinical practice, and long-term follow-up should be conducted.

RBD often appears in patients who have had Parkinson's disease for a long time and who exhibit severe symptoms [42, 43]. In addition, it is known to occur with high frequency in elderly patients [42, 43]. However, the intensity of REM sleep behavioral symptoms may not be obvious in elderly patients because it tends to change as the underlying neurodegenerative changes progress [44]. In addition, the symptoms of RBD were determined by interviewing family members of patients with Parkinson's disease [45].

In addition, this procedure is used to determine whether a detailed examination should be performed (such as polysomnography) [45]. If symptoms are mild, it can be difficult for family members to clearly recognize them. Therefore, screening for polysomnography by age alone is limited.

According to the results of this study, tests that focus on motor dysfunction, such as the exercise score of UPDRS and the total score of UPDRS, as well as cognitive screening tests such as the MoCA score and MMSE score, can more accurately predict the RBD high-risk population of elderly PD patients. However, more epidemiological studies should be

conducted to verify that aging and neuropsychological characteristics are important predictors of distinguishing AD from PD. Although it has more predictive power than traditional regression models (such as logistic regression), one disadvantage of machine learning is that it cannot explain the results. Therefore, future studies should adopt a hybrid approach, combining random forest with other learning models that has a high predictive power and can interpret the results.

Another finding of this study was that RF had better predictive performance than regression trees, as well as traditional logistic statistical techniques such as a discriminant analysis and a regression analysis. The results of this study are consistent with previous studies [46], which used RF to predict the high-risk population of cognitive impairment in the elderly population. It has been considered that the prediction performance of RF is superior to that of traditional statistical techniques (such as logistic regression analysis) or regression trees because RF is based on bootstrap aggregation (bagging) and generates diversified trees from multiple bootstrap samples. In other words, regression trees are at risk of overfitting because even an outlier is highly likely to be constructed as a node without exception, whereas RF can prevent overfitting because it generates multiple guided samples and has a better prediction accuracy than decision trees [47]. RF shows high predictive performance even when using unbalanced data, such as disease data for binomial classification [46]. Conversely, RF is a very cost-effective algorithm that can be quickly learned with little effect of outliers [48].

In particular, RF is suitable for analyzing big data, which is an advantage due to the nature of the algorithm. One of the most challenging tasks in applying machine learning techniques to the big data analysis environment is that data and models must be divided for model construction to deal with differences. RF is highly compatible with big data because its original algorithm was designed to appropriately divide data and input variables, to learn multiple decision trees and to later combine them. Therefore, RF can provide a better accuracy and sensitivity than regression trees or logistic statistics when analyzing multiple independent variables, such as disease big data, or when developing a model that predicts target variables by using data containing outliers.

## Strengths and limitations

The significance of this study is that it builds a model by using sociodemographic characteristics and disease examination data (evaluations of motor function, evaluations of nonmotor symptoms and evaluations of medication) and tests the predictive performance of machine learning techniques based on classification algorithms. The limitations of this study are as follows. First, various ways to improve RF performance should be considered in the future. RF performance can be related to the optimization of design parameters, such as Ntree or Mtry. However, the parameter optimization of Ntree cannot be considered in this study. Future research may require sensitivity analyses of RF parameters and the optimization of RF parameters. Second, it is necessary to consider multicollinearity more carefully when considering a large number of input variables and/or especially when applying statistical models. However, multicollinearity was not considered in this study. Therefore, future studies should consider taking multicollinearity into account when developing prediction models. Third, this study did not consider how to deal with classifier selection and data imbalance based on diversity and accuracy. In particular, due to the fact that this study did not observe data imbalance of target variables, it was not possible to determine the performance of machine learning in solving imbalance problems. Future studies need to test the predictive performance of machine learning for unbalanced data through classification algorithms. Fourth, this study did not investigate the daily habits of PD patients, such as smoking and drinking. Due to the fact that

the daily habits of Parkinson's disease may be associated with RBD, future research needs to develop a predictive model that considers daily habits.

## Conclusion

In conclusion, the incidence of Parkinson's disease complicated with RBD is high, and the course of the disease is longer. Motor disorders and motor complications were more serious; additionally, there were more nonmotor symptoms, including effects on overall sleep quality, autonomic symptoms, cognitive impairment and dopaminergic drug dosage. In this study, sociodemographic characteristics and disease test data (motor function evaluations, nonmotor symptom evaluations and medication evaluations) were analyzed, and the prediction performance of random forest was better than that of decision tree and traditional logistic regression analysis. The model that was established in this study to predict PD-RBD will be helpful in screening patients who should receive a detailed diagnosis. Based on the results of this study, further research is needed to develop a more advanced RF model that fully considers the multicollinearity of input variables and problems associated with data imbalance.

## Acknowledgments

We are very grateful for the help provided by the clinical staff, as well as to all of the patients and their families for participating in the study.

## Author Contributions

**Conceptualization:** Wu Chong-Wen, E. Xu.

**Data curation:** Li Sha-Sha, E. Xu.

**Investigation:** Wu Chong-Wen, Li Sha-Sha.

**Methodology:** E. Xu.

**Validation:** Li Sha-Sha.

**Writing – original draft:** Wu Chong-Wen, Li Sha-Sha, E. Xu.

**Writing – review & editing:** E. Xu.

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
