## [Decision Letter · Decision Letter 0]

2 Feb 2022

PONE-D-21-37123Predictors of rapid eye movement sleep behavior disorder in patients with Parkinson's disease based on random forest and decision treePLOS ONE

Dear Dr. Xu,

Thank you for submitting your manuscript to PLOS ONE. After careful consideration, we feel that it has merit but does not fully meet PLOS ONE’s publication criteria as it currently stands. Therefore, we invite you to submit a revised version of the manuscript that addresses the points raised during the review process.

The paper definitely needs English editing. Focus should also be made on using the same abbreviations everywhere in text (for example for PD without RBD: NO P-RBD, NP-RBD, PD-NRBD, PD-nRBD; which abbreviation should be used?)

References should be included everywhere in text, where needed (for example the rating scales for motor and nonmotor symptoms in Materials and Methods)

In Materials and Methods it is mentioned: ‘A total of 102 patients with Parkinson's disease who were admitted to…’and in the rest of the paper 169 patients are mentioned. What do the 102 represent?

The conclusions should be rephrased and presented more clearly in relation to the results.

We look forward to receiving your revised manuscript.

Kind regards,

Federica Provini

Academic Editor

PLOS ONE

2. Please amend your current ethics statement to address the following concerns: a) Did participants provide their written or verbal informed consent to participate in this study? b) If consent was verbal, please explain i) why written consent was not obtained, ii) how you documented participant consent, and iii) whether the ethics committees/IRB approved this consent procedure.

 “no”

“no”

Reviewer Comments:

Review Comments to the Author:

Reviewer #1:

The paper focuses on RBD in Patients with Parkinson’s Disease, especially in finding predicting factors for the development of RBD in PD Patients. It is an important issue, since RBD can be considered a preclinical marker for neurodegeneration and it has already been showed that PD patients with RBD have more severe motor and cognitive symptoms.

The authors compare three statistical methods for evaluating predictors for RBD in PD Patients. The statistical methods are extensively explained.

The paper can be published after some revisions.

The paper definitely needs English editing. Focus should also be made on using the same abbreviations everywhere in text (for example for PD without RBD: NO P-RBD, NP-RBD, PD-NRBD, PD-nRBD; which abbreviation should be used?)

References should be included everywhere in text, where needed (for example the rating scales for motor and nonmotor symptoms in Materials and Methods)

In Materials and Methods it is mentioned: ‘A total of 102 patients with Parkinson's disease who were admitted to…’and in the rest of the paper 169 patients are mentioned. What do the 102 represent?

The conclusions should be rephrased and presented more clearly in relation to the results.

---

## [Author Response · Author response to Decision Letter 0]

5 Mar 2022

Dear Editors and Reviewers,

On behalf of all co-authors, we appreciate you very much for the positive and constructive feedback on our manuscript entitled “Predictors of rapid eye movement sleep behavior disorder in patients with Parkinson's disease based on random forest and decision tree”. (ID:PONE-D-21-37123). We have studied reviewers’ comments carefully and have made revisions to the manuscript based on your comments. Below are details of how I have addressed the point raised. All of the changes have been highlight.

Responds to the reviewer’s comments:

1.The paper definitely needs English editing.

Responds: We are very sorry for the poor writing. The manuscript has been polished by the native speaker.

2.Focus should also be made on using the same abbreviations everywhere in text (for example for PD without RBD: NO P-RBD, NP-RBD, PD-NRBD, PD-nRBD; which

abbreviation should be used?)

Responds: We are so sorry for the informal spelling and we have checked all the abbreviations in the manuscript.

3.References should be included everywhere in text, where needed (for example the rating scales for motor and nonmotor symptoms in Materials and Methods)

Responds: Thank you so much for your comments, we have added related references in the manuscript.

4.In Materials and Methods it is mentioned: ‘A total of 102 patients with Parkinson's disease who were admitted to…’and in the rest of the paper 169 patients are mentioned. What do the 102 represent?

Responds: We are so sorry for the spelling mistake. It should be 169 patients.

5.The conclusions should be rephrased and presented more clearly in relation to the results.

Responds: Considering your valuable suggestion, we have rewritten the conclusion to make it more clear.

---

## [Decision Letter · Decision Letter 1]

30 Mar 2022

PONE-D-21-37123R1Predictors of rapid eye movement sleep behavior disorder in patients with Parkinson's disease based on random forest and decision treePLOS ONE

Dear Dr. xu,

Thank you for submitting your manuscript to PLOS ONE. After careful consideration, we feel that it has merit but does not fully meet PLOS ONE’s publication criteria as it currently stands. Therefore, we invite you to submit a revised version of the manuscript that addresses the points raised during the review process (please find in the attached file.

We look forward to receiving your revised manuscript.

Kind regards,

Federica Provini

Academic Editor

PLOS ONE

Journal Requirements:

Additional Editor Comments (if provided):

Reviewers' comments:

Reviewer's Responses to Questions

**Comments to the Author**

1. If the authors have adequately addressed your comments raised in a previous round of review and you feel that this manuscript is now acceptable for publication, you may indicate that here to bypass the “Comments to the Author” section, enter your conflict of interest statement in the “Confidential to Editor” section, and submit your "Accept" recommendation.

Reviewer #1: All comments have been addressed

2. Is the manuscript technically sound, and do the data support the conclusions?

Reviewer #1: Yes

3. Has the statistical analysis been performed appropriately and rigorously? 

Reviewer #1: Yes

4. Have the authors made all data underlying the findings in their manuscript fully available?

Reviewer #1: Yes

5. Is the manuscript presented in an intelligible fashion and written in standard English?

Reviewer #1: Yes

6. Review Comments to the Author

Reviewer #1: Thank you for answering all the questions. Some minimal writing improvements are needed (marked with yellow in the PDF file). Otherwise no further changes needed.

7. PLOS authors have the option to publish the peer review history of their article (what does this mean?). If published, this will include your full peer review and any attached files.

Reviewer #1: No

---

## [Author Response · Author response to Decision Letter 1]

3 Apr 2022

The response to reviewers has been uploaded.

---

## [Editor Report · Decision Letter 2]

20 May 2022

Predictors of rapid eye movement sleep behavior disorder in patients with Parkinson's disease based on random forest and decision tree

PONE-D-21-37123R2

Dear Dr. Xu,

We’re pleased to inform you that your manuscript has been judged scientifically suitable for publication and will be formally accepted for publication once it meets all outstanding technical requirements.

Kind regards,

Federica Provini

Academic Editor

PLOS ONE

---

## [Editor Report · Acceptance letter]

26 May 2022

PONE-D-21-37123R2 

Predictors of rapid eye movement sleep behavior disorder in patients with Parkinson's disease based on random forest and decision tree 

Dear Dr. Xu:

I'm pleased to inform you that your manuscript has been deemed suitable for publication in PLOS ONE. Congratulations! Your manuscript is now with our production department. 

Kind regards, 

on behalf of

Dr. Federica Provini 

Academic Editor

PLOS ONE